# Influence of Stress Severity on Contextual Fear Extinction and Avoidance in a Posttraumatic-like Mouse Model

**DOI:** 10.3390/brainsci14040311

**Published:** 2024-03-26

**Authors:** Noémie Eyraud, Solal Bloch, Bruno Brizard, Laurane Pena, Antoine Tharsis, Alexandre Surget, Wissam El-Hage, Catherine Belzung

**Affiliations:** 1Institut National de la Santé et de la Recherche Médicale (INSERM), Imaging Brain & Neuropsychiatry iBraiN U1253, Université de Tours, 37032 Tours, France; 2Pôle de Psychiatrie et d’Addictologie, Centre Hospitalier Régional Universitaire de Tours, 37000 Tours, France

**Keywords:** posttraumatic stress disorder, animal model, contextual fear conditioning, fear extinction, stress severity

## Abstract

Posttraumatic stress disorder (PTSD) is a widespread fear-related psychiatric affection associated with fear extinction impairments and important avoidance behaviors. Trauma-related exposure therapy is the current first-hand treatment for PTSD, yet it needs to be improved to shorten the time necessary to reach remission and increase responsiveness. Additional studies to decipher the neurobiological bases of extinction and effects on PTSD-like symptoms could therefore be of use. However, a PTSD-like animal model exhibiting pronounced PTSD-related phenotypes even after an extinction training directly linked to the fearful event is necessary. Thus, using a contextual fear conditioning model of PTSD, we increased the severity of stress during conditioning to search for effects on extinction acquisition and on pre- and post-extinction behaviors. During conditioning, mice received either two or four electrical shocks while a control group was constituted of mice only exposed to the context. Stressed mice exhibited important fear generalization, high fear reaction to the context and selective avoidance of a contextual reminder even after the extinction protocol. Increasing the number of footshocks did not induce major changes on these behaviors.

## 1. Introduction

Witnessing or experiencing a traumatic event can lead to the development of posttraumatic stress disorder (PTSD), which is prevalent in 10 to 30% of the exposed population [1]. More generally, about 3.9% of the population worldwide is affected by this psychiatric disorder [2]. To date, trauma-focused exposure therapy is the best first-line care offered to manage symptoms and improve quality of life [3,4]. Nevertheless, it requires patients repeatedly confront fear-evoking memories of their trauma or the stimuli usually avoided, and this often-distressing procedure requires numerous sessions, involves a high level of dropout and offers no guarantee of remission [5,6,7]. Additionally, only a few antidepressant medications can be prescribed, especially selective serotonin reuptake inhibitors (SSRIs) [7,8]. However, even when combined with psychotherapy, the therapeutic outcomes are relatively weak and not as recommended as psychotherapy alone [7,9]. Therefore, there is a pressing need to discover faster and more efficient methods to treat the condition and alleviate patients from the burden of lengthy, distressing or ineffective treatments. While some have attempted to develop new pharmacological treatments (e.g., propranolol, D-cycloserine, oxytocin…) [10], others have explored non-pharmacological and non-invasive methods to alleviate brain activity dysregulations found in PTSD. Numerous studies have thus focused on developing and adapting neurostimulation techniques to treat PTSD and augment the effects of psychotherapies [11,12,13,14]. Although the results are promising, discrepancies in outcomes and methodologies hinder drawing definitive conclusions about the efficacy of these methods, particularly repeated neurostimulation. Despite the challenge of fully capturing all dimensions of the disease and its complexity in animal models, there is a genuine need for additional preclinical studies to systematically explore new therapeutic avenues and elucidate the mechanisms underlying successes or failures of treatments. This is why we aimed to refine and adjust a rodent model to better represent the disorder and simulate exposure therapy.

PTSD has often been theorized as the formation of an abnormal and persistent fear memory [1,15]. Fear memory formation, consolidation and extinction processes have been extensively studied in humans and rodents through fear conditioning. It is a Pavlovian form of learning in which an aversive stimulus (e.g., an electrical shock) is taught to be associated with a neutral cue that is a sound, a light or even a context (CS—conditioned stimuli). Cues representing these CS are then centralized and processed to generate a memory of the event with appropriate fear responses (CRs—conditioned responses) [16]. Once the memory is properly stored, repeated exposure to internal or external cues allows one to apprehend the CS without the expression of CRs in the future. This phenomenon is called extinction, an inhibitory form of learning that requires the ventromedial prefrontal cortex (vmPFC) [16,17,18]. Using that definition of fear memory, when a person faces a fearful event, the most salient surrounding elements become aversive. However, over time, these salient cues can be interpreted as safe again. In PTSD patients, the fearful event was extremely traumatic, leading to intense fear reactions such as intrusive memories, flashbacks, vivid nightmares and avoidance behaviors that last for at least a month [19]. Therefore, this disorder has often been associated with failure in extinction mechanisms [15,20]. Moreover, as mentioned earlier, extinction learning relies on the vmPFC. Yet, hypoactivity in this region has frequently been uncovered in PTSD patients, which has been correlated with extinction deficits [21,22,23]. Additionally, extinction learning has been conceptualized as a model of exposure therapy [24]. Hence, a clear animal model of PTSD, incorporating an extinction protocol directly linked to the fearful experience but avoiding complete fear extinction, could aid in studying ways to enhance extinction-based therapies and understand associated mechanisms.

Numerous animal models eliciting PTSD-like symptoms have been developed [25,26]. Regarding fear extinction, the model developed by Siegmund and Wotjak, which is based on contextual fear conditioning (CFC), appears to be particularly appropriate [27]. It relies on the association of an unknown context with an intense aversive stimulus (2 s long 1.5 mA electrical shock) and induces well-defined behaviors associated with PTSD symptoms. Indeed, this model has reached face validity since it reproduces core symptoms of the disorder [27,28]: the exposed rodents exhibit increased anxiety, intense fear when re-exposed to the context, avoidance behaviors, enhanced startle responses, sleep disturbances and social avoidance [27,29,30,31,32,33]. Animals also express fear behaviors when exposed to other contexts not associated with the fearful event (overgeneralization of contextual fear) [32,33]. Paroxetine and fluoxetine, two SSRIs used in the treatment of PTSD, significantly reduce model-related symptoms and behaviors [27,29,33,34]. Therefore, it also met criteria in terms of predictive validity by responding to common pharmacological treatments [27,28]. Furthermore, extinction learning can easily be studied by re-exposing the animals to the conditioned context several times [32]. But long-term outcomes of the model need to be further investigated after the extinction acquisition. Few studies have investigated the repercussion of contextual fear extinction on PTSD features induced by the model. Golub et al. (2009) [32] showed that conducting extinction training 28 days after the aversive event had a positive effect on fear generalization but not on hyperarousal. On another note, Pamplona et al. (2011) [33] investigated the impact of the same extinction training on generalized avoidance behaviors, which was reduced compared to a condition without extinction (though not fully alleviated). However, besides fear generalization, few PTSD-like symptoms have been observed to manifest following extinction training. If future studies aim to explore the advantages of augmenting extinction learning, a model showing symptoms after extinction training may prove beneficial.

PTSD has frequently been conceptualized as a robust fear conditioning learning that is resistant to extinction-inhibitory processes. However, PTSD is also thought to be influenced by the severity of the trauma and the level of stimulation during fear acquisition [9]. Therefore, the greater the fear and stress levels experienced during the event, the higher the risk of developing PTSD. Using this PTSD-like model based on CFC, very few studies have investigated the impact of varying the severity of the aversive stimuli, during conditioning, on the strength and the persistence of contextual fear expression and extinction learning, as well as on other behaviors representing PTSD symptoms. A previous study in rats already showed that increasing the number of footshocks during conditioning can strengthen the expression of contextual fear and fear generalization a month later [35]. In this study, we chose to increase the number of footshocks in the CFC mouse model of PTSD to dissect its repercussions on CFC expression but also on extinction memory acquisition and retrieval and other PTSD-like symptoms including avoidance behaviors when facing a novel environment and/or a contextual reminder. We hypothesized that increasing the severity of stress could lead to less sensitivity to extinction as well as more stable PTSD-like symptoms expressed before and after extinction learning.

## 2. Materials and Methods

### 2.1. Animals

We purchased 58 Male C57BL6NRj mice (Janvier labs, Le Genest-Saint-Isle, France) at 8 weeks of age. They were individually housed in type II non-ventilated cages at a room temperature of approximately 20–22 °C. Food and water were accessible ad libitum. Experiments were conducted two weeks after arrival and during the dark phase of a 12:12 light/dark cycle.

### 2.2. Experimental Design

The experiment, lasting 37 days (Figure 1), was designed to evaluate the long-term effects of the severity of an aversive stimulus used for CFC on PTSD-like symptoms before, during and after an extinction protocol. Mice were allocated to three different groups: one group received two electrical shocks as the unconditioned stimulus (*n* = 20; PTSD2), a second group received four electrical shocks in the same context (*n* = 20; PTSD4) and a control group received no electrical shocks (*n* = 18; CTRL). Mice of the same group were placed in pairs into the context during CFC. Three weeks later, when mice were placed back into this identical context to induce fear extinction, they were paired with the same mouse as during conditioning. Before the extinction protocol, cognition was evaluated using the Novel Object Recognition test (NOR), and the avoidance of a contextual cue was tested through an avoidance test. After extinction learning, we evaluated extinction retrieval as well as avoidance, but also anxiety behaviors using the light and dark box test (LDB). The experiment was initially conducted with 28 mice (first batch) and repeated with an additional 30 mice (second batch) with a two-week interval. Only the second batch underwent investigation for avoidance and anxiety behaviors following extinction learning, with 10 mice per group. The first batch was euthanized for a histological investigation, which was unrelated to the present study. All procedures were in accordance with the Directive 2010/63/EU guidelines and approved by the French ethics committee for animal experimentation n°019 (Project n°2021111812158033).

### 2.3. Contextual Fear Conditioning

The CFC was induced by high-intensity footshocks to model PTSD, a procedure shown to engender behavioral responses reminiscent of PTSD symptoms [27]. Mice were placed in a fear conditioning apparatus (Noldus/Ugo Basile Fear conditioning setup) that is a sound-attenuation chamber containing a cage (26 × 26 × 36 cm(h)cm) with a grid floor allowing footshock delivery. The cage was divided into two identical compartments (26 × 12.25 × 36 (h)cm) thanks to an opaque Plexiglas separator. The conditioning apparatus was unlit. Two identical objects (a glass prism 2.5 × 2.5 × 2.5 × 6.5 cm) were placed into the context, one on each side of the separator. Walls, the separator, the grid and the objects were cleaned between each mouse with a solution of Eugenol (Acros Organics, CAS: 97-53-0) diluted in water at 8.5 mM. Two mice were placed at the entrance of the cage, each on their own side, with the separator preventing them from seeing each other. They were allowed to explore the context for 190 s before a footshock-delivery period (two or four consecutive footshocks) and for 60 additional seconds after the last footshock. Two-second-long shocks (=1.5 mA) were delivered through the grid separated by six-second-long inter-shock intervals. Thus, the procedure lasted 4 min and 20 s for the PTSD2 group and 4 min 36 s for the PTSD4 group. CTRL mice were allowed to explore the context for 4 min 36 s. Timing of the test and of the inter-shock intervals were controlled by EthoVision (EthoVision XT, 16.0, Noldus IT, Wageningen, The Netherlands).

### 2.4. Behavioral Tests

#### 2.4.1. Novel Object Recognition Test (NOR)

This test consisted of three phases: a habituation phase, a learning/familiarization phase, and a testing phase, following the guidelines by Leger et al. (2013) [36]:Habituation phase (day 16 or 17): Mice were placed at the center of the apparatus (a cylindric open field (OF), 38 × 30 (h)cm) lit with a very low white light (30 lux at the center) and were allowed to explore for 10 min.Learning phase: The next day (Day 17 or Day 18), two identical objects (either two moon- or butterfly-shaped plastic toys) were stuck on the wall of the OF at opposite ends about 3 cm from the ground. Mice were left in the apparatus until they had explored the objects for 20 s (total time spent smelling both objects) or had been in the OF for a maximum of 10 min.Testing phase: Exactly one hour after the learning phase, mice were put back into the OF but one of the two objects had been swapped with a new one identical in terms of size and material but with a different shape and color (e.g., Two moons during the learning phase compared to a moon (known) and a butterfly (new) during the testing phase). The familiar object and the new object were placed on the wall at the same distance as during the learning phase. Mice were left in the apparatus for 10 min and the time spent exploring each object was measured manually.

The distance moved in the OF was measured with EthoVision during the habituation and the testing phase.

Mice were habituated to the room for 30 min before each phase.

#### 2.4.2. Avoidance Test

The avoidance test was inspired by the conditioned odor avoidance test from Pamplona et al. (2011) [33]. It was performed in a three-chamber apparatus (60 × 42 × 22 (h)cm) divided into three chambers (20 × 42 × 22 (h)each) communicated by small openings that could be closed by small doors, and under a very low red light. The object present during conditioning (the glass prism—2.5 × 2.5 × 2.5 × 6.5 cm—which we referred to as the reminder) was exhibited in one of the chambers adjacent to central chamber and cleaned with the same Eugenol solution. In the opposite chamber, cleaned with water, a neutral object (a flat and large metallic ring—6 × 1 (h)cm with a 3-cm-large hole in the middle), to which animals were habituated in their home cage before the test, was first displayed (4 min and 36 s on day 20). The odor had previously been tested on naïve mice to make sure that there was no spontaneous preferences or avoidance compared to water (Appendix A). The apparatus was cleaned with the usual cleaning spray between each mouse. Mice were individually placed in the center chamber with entrances to the side chambers closed. Doors were opened after a period of three minutes (habituation) and mice were able to explore the two chambers containing the objects for a total of ten minutes. The time spent in each chamber and the distance moved were measured by EthoVision, while the time spent exploring each object was measured using the manual scoring function of the software. This test was executed before (Day 21) and after (Day 35) the extinction protocol (see below Section 2.4.3).

The mice were transferred from their housing room to the behavioral testing room 30 min before initiating the avoidance tests, allowing them to acclimate to the new environment.

#### 2.4.3. Re-Exposure and Extinction

Extinction learning: Mice were re-exposed to the conditioning context for 20 min, once a day over three consecutive days (Day 23 to Day 25). The context was identical to that during conditioning (room, mice pairing, sound, light, smell and with the object on each side of the separator) but mice did not receive any footshock. Freezing was measured by EthoVision and analyzed as a percentage of time they froze over periods of 5 min.

Extinction retrieval: Three days after the last extinction day (Day 28), learning of extinction was tested by re-exposing mice to the context once more for only 5 min. The percentage of time frozen over the entire session was measured by EthoVision. The percentage of extinction for each mouse was calculated by using the following formula:$$\%\;\mathrm{of}\;\mathrm{extinction}\;=\;\frac{\mathrm{freezing}\;\mathrm{during}\;5\;\mathrm{first}\;\min\;\mathrm{of}\;\mathrm{the}\;1\mathrm{st}\;\mathrm{re}-\mathrm{exposure}\;-\;\mathrm{freezing}\;\mathrm{during}\;\mathrm{retrieval}}{\mathrm{freezing}\;\mathrm{during}\;5\;\mathrm{first}\;\min\;\mathrm{of}\;\mathrm{the}\;1\mathrm{st}\;\mathrm{re}-\mathrm{exposure}}\times 100$$

Z-scores were calculated using the mean and the standard deviation of both PTSD2 and PTSD4 mice gathered. For Z re-exposure, the individual mean of the freezing over the 20 min of re-exposure for each mouse was used as the X.
$$µ\;Z\;\mathrm{re}-\mathrm{exposure}\;=\;\frac{Z\;1\mathrm{st}\;\mathrm{re}-\mathrm{exposure}\;+\;Z\;2\mathrm{nd}\;\mathrm{re}-\mathrm{exposure}\;+\;Z\;3\mathrm{rd}\;\mathrm{re}-\mathrm{exposure}}{3}$$

#### 2.4.4. Light and Dark Box (LDB)

The apparatus was composed of two boxes: a light box (20 × 20 × 15 (h)cm), with transparent walls exposed to a 250-lux white light, and a dark box of the same dimensions but with opaque walls and a roof to block out the light. A small opening enabled the mice to freely explore one box or the other. Animals were placed into the dark box and the test lasted 5 min. The total time spent into the light box and the latency to enter it were measured manually. An animal was considered as in the light box when all four paws were in the box. This test took place after extinction (Day 36) to have an anxiety index. Thirty minutes before the test, all mice were transferred to the behavioral testing room to allow them to acclimate to the environment and to mitigate any stress biases associated with the relocation.

### 2.5. Statistical Analysis

For most of the behavioral tests, only the group factor intervened (CTRL vs. PTSD2 vs. PTSD4). Normality (Shapiro–Wilk) and sphericity (Bartlett) conditions were checked. Behavioral data were analyzed using ordinary one-way ANOVA followed by Tukey’s post hoc test (only when *p* < 0.05) or the Kruskal–Wallis test followed by Dunn post hoc tests when a non-parametric test was required. For the NOR test, avoidance test and extinction retrieval test, data were compared between groups (stress factor) but also between objects within the same group (object factor as a repeated measure) or freezing during retrieval compared to the first five minutes of the first re-exposure (day factor as a repeated measure). Thus, a two-way repeated measure ANOVA was used, followed by Tukey’s post hoc test (independent factors) and the Sidak test (repeated factors) when needed. To compare NOR data to 50%, a one-sample *t*-test was used. All analyses were performed on GraphPad Prism (Version 8.0.1, GraphPad Software, Boston, MA, USA). Finally, the evolution of freezing over the different re-exposures during the extinction learning was analyzed using a general linear model (GLM) considering three factors: the stress factor (unpaired: CTRL vs. PTSD2 vs. PTSD4), the day factor (repeated measure: the first vs. the second vs. the third re-exposure) and the time factor within each day (5 min vs. 10 min vs. 15 min vs. 20 min within re-exposures). A Khi^2^ was used to compare the proportion of PTSD2 mice with the lowest extinction to the proportion of the PTSD4 group with a comparable extinction. The GLM analysis was performed with Statistica 64 (version 12, Stat Soft. Inc., Las Vegas, NV, USA). All data are presented as mean ± standard error to the mean (SEM). Detailed statistics can be found in Appendix A.

## 3. Results

### 3.1. Strengthening the Severity of Stress during Conditioning Did Not Drastically Influence Fear Response to the Context nor Extinction Learning

The extinction protocol lasted 20 min per day over three consecutive days. The evolution of fear over the sessions was assessed by measuring the time the animal froze over successive 5 min long time bouts and expressed in the percentage of time spent freezing (Figure 2A). Three days later, mice were re-exposed one last time for a 5 min long period to test the extinction retrieval.

First, the results showed that three weeks after fear conditioning, mice expressed important freezing responses to the context compared to CTRL (*p* < 0.0001 for PTSD group vs. CTRL) without differences caused by stress severity (*p* = 0.932). The same result was found at the second and third re-exposures (see Table 1). However, when repeatedly re-exposing the animals to the context, freezing decreased over days (day 1 > day 2 and day 2 > day 3 for both PTSD groups; see Table 1) underlying the expression of the extinction learning processes. To see whether extinction learning was acquired, the freezing level during extinction retrieval was compared to the freezing level before the beginning of extinction learning, so during the first five minutes of the first re-exposure. While both PTSD groups froze significantly more than CTRL (two-way ANOVA interaction: F(2, 55) = 14.34; *p* < 0.0001; Tukey’s post hoc: PTSD2 or PTSD4 vs. CTRL *p* < 0.0001), they also froze significantly less than during the first re-exposure (Sidak’s post hoc: *p* < 0.0001 for both groups). Therefore, the extinction protocol induced extinction learning for the PTSD mice from both groups. Nonetheless, fear expression was still significantly higher in both PTSD groups than in CTRL.

Then, we examined if extinction was acquired differently according to the stress severity. Hence, the percentage of extinction was calculated, representing the decrease in fear between the retrieval and the first re-exposure (Figure 2B). The percentage of extinction did not appear modified between the groups (*p* = 0.361). Finally, when computing Z-scores during re-exposures in relation to the Z-score of retrieval (see Section 2), we can obtain an overview of the population repartition according to how well each mouse acquired extinction learning over the re-exposures compared to the entire population (µ Zre-exposure) and how stable this extinction was over time (Zretrieval; Figure 2C). This repetition showed that 35% of mice from the PTSD2 group were considered to have a weaker fear extinction (Z-scores > 0, red right top corner) compared to 45% of the PTSD4 group. Nevertheless, these two proportions were not significantly different (Khi^2^, *p* = 0.52).

### 3.2. A Strong Contextual Fear Conditioning Induced PTSD-like Behaviors

After fear conditioning (day 0), isolated mice were left in their home cage for 16 days. The first behavioral test performed was a NOR test to evaluate learning and memory deficits. This test started with a habituation phase to the arena, a dimly lit open field that allowed us to measure locomotion (Figure 3A). Interestingly, we observed that both PTSD groups traveled less distance in this new environment compared to CTRL (one-way ANOVA: F(2, 55) = 10.58; *p* = 0.0001; Tukey’s post hoc: CTRL vs. PTSD2, *p* = 0.0004; CTRL vs. PTSD4, *p* = 0.0007). This distance moved during the habituation phase of the NOR was negatively correlated with the expression of freezing during the first five minutes of the first re-exposure (day 23) only for the PTSD4 group (Pearson: r = −0.542, *p* = 0.0135). The correlation indicated that the mice that froze the most when first re-exposed to the conditioning context were also the mice that traveled the least distance in an open field (Figure 3B). During the learning phase of the NOR, mice were exposed to two identical objects until they became familiarized (20 s of exploration or else 10 m in the apparatus). The CTRL group smelled the objects for an average of 19.28 s (±0.65 s), the PTSD2 group for 17.25 ± 1.08 s and 16.65 ± 1.31 s for the PTSD4 group. Additionally, the time they took to reach this object familiarization was increased in the two PTSD groups compared to CTRL (Figure 3C); Kruskal–Wallis: H(2, 55) = 11.39, *p* = 0.0034; Tukey’s post hoc: CTRL vs. PTSD2, *p* = 0.0042; CTRL vs. PTSD4, *p* = 0.0277). During the testing phase of the NOR, while the CTRL mice and PTSD2 mice explored the new object for the same amount of time than the known object (=50%, Figure 3E), the PTSD4 mice explored it less (<50%, one-sample *t*-test, *t* = 3.539, *p* = 0.0022). Moreover, they explored it significantly less than the PTSD2 mice (one-way ANOVA: F(2, 55) = 4.113, *p* = 0.0216; Tukey’s post hoc, *p* = 0.0345) and tended to explore it less than the CTRL mice (*p* = 0.0545). Although, this decrease in the object exploration was not linked to an overall decrease in mobility since the three groups did not differ in the distance moved during the test (Figure 3D); one-way ANOVA: F(2, 55) = 2.221, *p* = 0.1182).

The avoidance test (day 21) was used to assess whether the different conditioning protocols resulted in mice avoiding a reminder of the aversive event (e.g., the object present in the context during conditioning) in the long run. First, the distance traveled within the arena during the test was lower in the two PTSD groups compared to the CTRL group (Figure 3F); Kruskal–Wallis: H(2, 55) = 10.21, *p* = 0.0061; Dunn’s post hoc: CTRL vs. PTSD2, *p* = 0.0133; CTRL vs. PTSD4, *p* = 0.0187). Then, we looked at the time spent exploring the reminder compared to the time spent exploring a neutral object (Figure 3G). The CTRL mice equally interacted with both objects (two-way ANOVA, object factor: F(1, 55) = 23.28, *p* < 0.0001; Sidak’s post hoc: *p* = 0.999). Nevertheless, even though the PTSD mice explored the objects less in general (two-way ANOVA, group factor: F(2, 55) = 18.67, *p* < 0.0001; Tukey’s post hoc: PTSD2 or 4 CTRL *p* < 0.0001), they interacted even less with the reminder (*p* < 0.0001 for both PTSD groups). This result was confirmed when looking at the time spent exploring the chambers containing the objects (Figure 3H). Indeed, the chamber containing the neutral object was explored the same amount by the PTSD mice than the CTRL mice (two-way ANOVA, group×chambers interaction: F(2, 55) = 12.83; *p* < 0.0001, Tukey’s post hoc: CTRL vs. PTSD2, *p* = 0.989; CTRL vs. PTSD4, *p* = 0.994). In contrast, they spent significantly less time within the chamber containing the reminder (*p* < 0.0001 for both), underlying a clear avoidance.

### 3.3. The Avoidance of the Contextual Reminder Was Maintained after Extinction

Thirty-five days after conditioning but also ten days after having followed a fear extinction protocol, mice went through the avoidance test once more and the behaviors were analyzed in the same manner starting with the distance travelled during the test (Figure 4A). As for the first avoidance test, the one-way ANOVA revealed differences between groups (F(2, 27) = 7.715, *p* = 0.0022), highlighting a lower exploration in the PTSD2 group compared to CTRL (Tukey’s post hoc: *p* = 0.0016) but not to PTSD4 (*p* = 0.2861). PTSD4 mice tended to travel less distance than CTRL without reaching significance (*p* = 0.0652). The time spent exploring the neutral object and the reminder was also compared between groups using a two-way RM ANOVA and revealed significant differences between groups (group × objects interaction: F(2, 27) = 6.899; *p* = 0.0038). PTSD2 mice explored the reminder significantly less compared to CTRL (Tukey’s post hoc: *p* = 0.0033) and the difference spent on the reminder tended to be lower than the time spent exploring the neutral object (Sidak’s post hoc: *p* = 0.0684). Furthermore, compared to CTRL, PTSD4 mice seemed to explore the reminder less (Tukey’s post hoc: *p* = 0.0870) and explored the neutral object more (*p* = 0.0337). More importantly, they explored the neutral object significantly more than the reminder (Sidak’s post hoc: *p* = 0.0141). Finally, analyses of the time spent in the chamber containing the reminder compared to the chamber containing the neutral object confirmed a difference between groups (group × objects interaction: F(2, 27) = 7.020; *p* = 0.0035). While CTRL spent the same time exploring both chambers (Sidak’s post hoc: *p* > 0.9999), the PTSD mice explored the chamber containing the reminder less compared to the one with the neutral object (PTSD2: *p* = 0.0002; PTSD4: *p* = 0.0006). Furthermore, PTSD mice spent less time within the chamber containing the reminder (Tukey’s post hoc: PTSD2, *p* = 0.0005; PTSD4, *p* = 0.0038) and tended to explore the chamber containing the neutral object more compared to CTRL (PTSD2, *p* = 0.0765; PTSD4, *p* = 0.0689). Taken together, these results confirm the maintenance of avoidance behaviors after a protocol of extinction.

At last, during the LDB, PTSD mice did not take longer to enter the light box (Kruskal–Wallis, H = 1.456, *p* = 0.4828), nor did they spend less time exploring the light box than CTRL mice (H = 2.093, *p* = 0.3512). Based on this test, PTSD mice did not exhibit increased anxiety-like behaviors compared to CTRL mice.

## 4. Discussion

In this CFC study on mice, our aim was to intensify the severity of the fearful event to investigate the consequences on contextual fear, extinction memory and additional PTSD-like phenotypes before and after extinction training. The results first showed that the model used led to high fear response when first re-exposed to the conditioned context, only partial extinction following multiple re-exposures, avoidance of a contextual reminder of the fearful event even days after the extinction training and lower exploration behaviors in unknown environments. Furthermore, increasing the severity of stress during conditioning did not result in markedly increased PTSD-related behaviors as expected since it did not amplify avoidance phenotypes, nor did it increase contextual fear generalization in unknown environments. Though, it did decrease interaction with the novel object in the NOR.

When first re-exposed to the context and before any extinction learning, both our conditioned groups froze in response to the context for the same amount of time. When the first re-exposure began, mice from both PTSD groups froze around 75% of the time at the beginning of the session, suggesting a significant association of the context with the fearful event. However, there was no dose–response effect due to the number of shocks. Previous studies in rats showed that one day after conditioning, freezing to the context was increased when several shocks had been applied compared to only one shock [35,37]. In the study of Poulos et al. (2016) [35], there was a dose effect when rats were re-exposed to the context the next day with a gradual level of freezing according to the number of shocks. But 28 days later, freezing increased with one or more shocks but not systematically between two and five shocks. Yet, Rau and Fanselow (2009) [37] found no difference in freezing levels after a day between four and fifteen shocks. This could imply that a maximum level of freezing (ceiling effect) is attained after two or more shocks, even weeks after the conditioning acquisition, when long-term memory has been fully stored and consolidated. However, extinction could still be differentially acquired, showing difficulties in expressing the inhibitory learning due to an over expressed conditioning memory. Our results showed that, during extinction learning, freezing decreased similarly. None of the groups had a more obvious extinction learning deficiency during acquisition. Plus, freezing during extinction retrieval was similar in both groups. In that regard, we also calculated the percentage of extinction (i.e., how much freezing decreased after the extinction learning according to freezing reaction before extinction), but no differences were highlighted between both conditioned groups. Surprisingly, while 35% of mice in the two-shock group displayed elevated fear levels during both extinction learning sessions and retrieval, a comparable freezing level was observed in 45% of mice in the four-shock group; however, these proportions did not show statistically significant differences. Hence, it appears that increasing the number of shocks did not affect extinction learning, nor did it enhance conditioning memory. There does not seem to be a differential interpretation of threat severity between two or four shocks. Nonetheless, fear expression remained high even after a three-day extinction protocol. Although the extinction training was not entirely ineffective, there is still room for improvement.

On another aspect, fear generalization is a common adaptative process necessary for safety learning by interpreting novel situations according to previous events [38]. However, in PTSD patients, there is an overgeneralization that makes them fear or avoid cues (e.g., a situation, a place, a person, a sound or something else) not necessarily related to the aversive event (DSM-5, Criterion C avoidance symptom; refs [19,38]). Such generalization was previously described in Siegmund and Wotjak’s PTSD-like CFC model, with the main difference with our study being that only one shock was applied during CFC acquisition. When exposing mice to contexts different from the conditioned one, increased freezing was observed [31,32,39]. Moreover, in a three-chamber context, a general avoidance of chambers containing either the smell used during conditioning or a different smell compared to a chamber containing the nest smell was reported [31,33].

In our study, mice received either two or four footshocks. Previously, a CFC study in rats by Poulos et al. (2016) [35] demonstrated that increasing the number of footshocks during conditioning increased fear generalization to other contexts. Here, we found an overall decrease in exploration in unknown contexts (i.e., habituation phase of the NOR and avoidance test) in both fear conditioned groups without a notable impact of increased number of footshocks. A lessened exploration in an open field after CFC (or CFC-based protocols) has previously been interpreted as increased fear generalization [40,41,42]. Thus, we interpreted reduced exploration as indicative of fear generalization, especially since in the PTSD group that received the highest number of shocks, the mice that traveled the least in the OF, were also the ones that exhibited the highest freezing response during the first re-exposure to the conditioned context. Additionally, a similar correlation was previously found to predict susceptibility to exhibit PTSD-like phenotypes in rats [43]. Moreover, during the NOR test, PTSD mice took longer to become familiarized to a new object than the controls. Then, during the testing phase of the NOR, mice that had received four shocks were the only ones to significantly avoid the unfamiliar object compared to the familiar one. All these results taken together suggest that our model induced fear generalization to novel contexts, which might have been accentuated by a more severe stress by generalizing fear to objects. Testing fear generalization in contexts slightly similar to the conditioned one could have helped distinguish associative from non-associative types of generalizations (e.g., using a context with a grid floor and another similar one in terms of size), potentially enhancing the assessment of overgeneralization by measuring freezing expression [31,32,33]. However, it is essential to note that not measuring the freezing response directly but rather exploring other behaviors should be more systematically considered, as emphasized by the results of this study. Freezing is indeed a specific indicator of fear but considering other less obvious behaviors might help prevent under-interpretation. Although, caution must be taken regarding the interpretation of the results of the NOR since the control mice interacted equally with the unknown object than with the familiar object. Indeed, the NOR test is a memory test based on the mice’s interest in novelty. The theory is that mice exhibiting functional memory should spend more time exploring the novel object compared to the familiar one [36,44,45]. Therefore, this absence of novelty preference in controls most probably underlines an absence of memory retention in our procedure and/or, conversely, an hypermnesia with novelty avoidance in more severely stressed mice.

Fear generalization after CFC has been shown to depend on the infralimbic cortex (IL—the rodent equivalent of the vmPFC) [46], just like the vmPFC seems implicated in fear generalization in humans [47]. Additionally, the IL plays a central role in extinction learning and expression [48]. Studies have revealed that CFC leads to a reduction in IL excitability [49]. Thus, fear generalization and only partial extinction learning may reflect a long-term impact of the stressful event on the IL functioning, potentially contributing to PTSD-like symptoms. Further exploration on the IL’s involvement in overgeneralization and specific avoidance can shed light on the efficacy of targeting fear extinction in PTSD treatment. Avoidance is a major symptom of PTSD. Patients actively avoid confrontation with anything that could remind them of the traumatic event, which reduces possibilities to extinguish fear responses [4,50]. The efficacity of therapy was found to be associated with reduced avoidance of trauma cues and it was argued to be a representative factor in therapy-related clinical improvement [4,51]. We observed a significant avoidance of a reminder of the conditioned context that persisted even after an extinction protocol, providing further evidence that extinction learning was incomplete. In addition to avoiding the reminder, mice avoided the chamber containing that reminder. Our protocol was inspired by the one from Pamplona et al. (2011) [33]. In their study, they used the conditioned odor as compared to a different smell never encountered and the smell of the animal’s nest. In that way, they found that mice exhibited specific avoidance of the conditioned odor two days after conditioning. Yet, differing from our results, this avoidance had spread to the unknown odor after 28 days of fear incubation, and hence was not specific to the conditioned odor anymore. Additionally, extinction training as well as previous habituation to the apparatus significantly improved the characterized avoidance behaviors. In our study, the reminder was the conditioned object associated with the conditioned smell (see Section 2), a more complex and salient cue than an odor alone. A general avoidance of both objects (reminder and neutral objects) by both PTSD groups was seen before the extinction protocol. At that point, animals had never been exposed to the apparatus and exhibited decreased exploration of this unknown context, which confirms the importance of habituation. Nonetheless, it did not prevent a specific avoidance of the reminder object compared to the neutral one. Therefore, this avoidance behavior does not indicate a generalization of fear to other contexts but a specific association of the fearful event with the contextual elements during conditioning, which could potentially be ameliorated through enhanced extinction learning. The specific avoidance of the reminder was also found 10 days after extinction training. To be noted, a smaller number of mice were tested after extinction than before extinction for the same avoidance test (*n* = 10 instead of *n* = 18 or 20 per group), which did not prevent us from uncovering avoidance of the reminder or the chamber containing the reminder, especially in the PTSD4 group.

This study has several limitations that should be acknowledged. Firstly, the most significant limitation is its restrictions to male subjects. PTSD has been found to occur twice as often in women compared in men. Thus, it is crucial to include females in preclinical and fundamental studies. However, it is worth noting that the type of traumatic event could also play a significant role. For instance, sexual assaults have been particularly associated with the development of PTSD, a type of trauma more often experienced by women [2,52]. Furthermore, a few studies on contextual fear conditioning have highlighted differences between males and females in the acquisition and expression of fear extinction, showing a higher level of freezing in males compared to females. However, this suggests that freezing might not be the sole representation of fear, and other behaviors must be defined and refined [53]. Additionally, this underscores the challenge of translating animal models to humans, despite the necessity of animal investigations to explore new avenues in clinical research. Moreover, post-extinction behaviors were assessed one month after CFC. One month is already a lengthy period for fear conditioning studies in rodents, which are typically limited to 28 days. Nonetheless, it could be valuable to continue this investigation at other time points to capture the long-term effects of extinction. Finally, a group subjected to fear conditioning with a protocol not expected to induce PTSD-like behaviors (e.g., a single 0.7 mA shock) could have been useful to discern what is attributable to fear learning and what can genuinely be associated with extreme fear consequences.

## 5. Conclusions

In summary, our study further underlines a possible ceiling effect of increasing the number of shocks on fear reaction to a conditioned context. Another way to increase stress severity is to increase shock intensity [27,54,55], although 1.5 mA is already strong and highly above the pain threshold of C57BL6N mice [27]. In our hands, increasing to four shocks did not drastically impact extinction learning nor other PTSD-like behaviors in comparison to two shocks. Interestingly, PTSD-like behaviors were maintained after extinction learning and obtained with a small number of mice per group. To conclude, this model could be useful to study new therapeutic approaches to enhance extinction learning, and to alleviate PTSD symptoms such as fear generalization and trauma-related avoidance. Indeed, we know that extinction learning mostly relies on the top-down control of the vmPFC over the amygdala. Using this model to study new therapeutic approaches that could aim to increase vmPFC activity (e.g., neurostimulation techniques [29,48,56,57]) or enhance fear extinction with molecules known to interact with this inhibitory learning (e.g., curcumin [58,59] or ketamine [60]) might help us find efficient protocols to accompany or even enhance trauma-focused psychotherapies.

## Figures and Tables

**Figure 1 brainsci-14-00311-f001:**
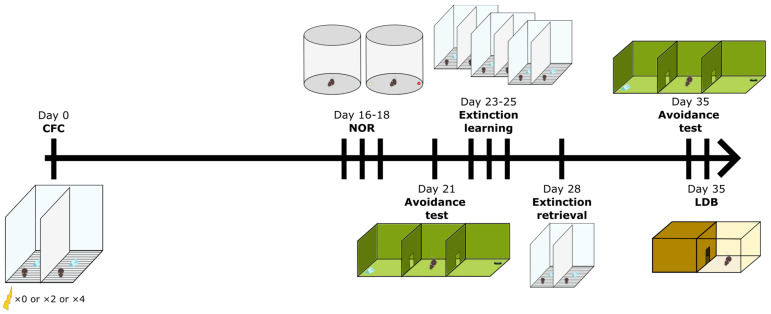
Timeline of behavioral tests performed by mice from all groups. Stressed mice were conditioned to a specific context by receiving either two footshocks (PTSD2 group, *n* = 20) or four footshocks (PTSD4 group, *n* = 20), while mice from the control group (CTRL, *n* = 18) were only exposed to the context without receiving any electrical shock. Behaviors mimicking PTSD symptoms were evaluated before and after a three-day extinction learning protocol (Days 23–25, 20 min re-exposures to the context without footshocks). Extinction retrieval (Day 28) corresponded to a five-minute-long exposure to the context without footshocks, three days after the extinction learning. The object present in the conditioning context was used as a reminder of the fearful event during the avoidance test (Day 21 and Day 35) and compared to a familiar but neutral object. Objects used during the Novel Object Recognition test (NOR) were never presented to the animals before nor reused for other tests. LDB stands for light and dark box test. CFC stands for contextual fear conditioning.

**Figure 2 brainsci-14-00311-f002:**
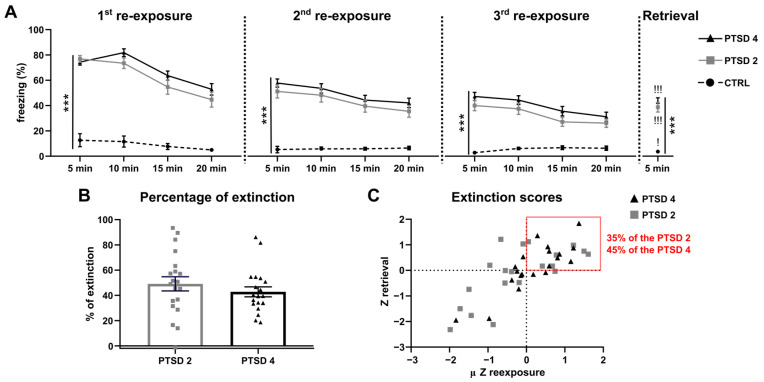
Repercussion of the number of footshocks on extinction learning and retrieval. (**A**) Freezing evolution over the three 20 min long sessions of extinction learning and during extinction retrieval. Conditioned mice froze significantly more than CTRL, regardless of the number of shocks they had received. (**B**) The percentage of extinction was calculated using the first five minutes of freezing at the first re-exposure. No significant difference was highlighted between the two conditioned groups. (**C**) To determine the portion of the population with the lowest extinction, Z-scores of re-exposures during extinction learning were calculated using the mean and standard deviation over both conditioned groups at once. The mean of the three Z-scores was then used (µ Z re-exposure). Z retrieval corresponds to the Z-score of freezing during retrieval over both conditioned populations undifferentiated. Forty-five percent of the mice from the PTSD4 group exhibited a low extinction and thirty-five percent of PTSD2 mice showed the same low extinction (mice from the top right corner in red). Results are represented by the mean ± SEM, *** *p* < 0.001 vs. CTRL; ! *p* < 0.05 and !!! *p* < 0.001 retrieval vs. first five minutes of the same group during the first re-exposure, control group (*n* = 18); PTSD2, PTSD-like group fear conditioned with two footshocks (*n* = 20); PTSD4, PTSD-like group fear conditioned with four footshocks (*n* = 20).

**Figure 3 brainsci-14-00311-f003:**
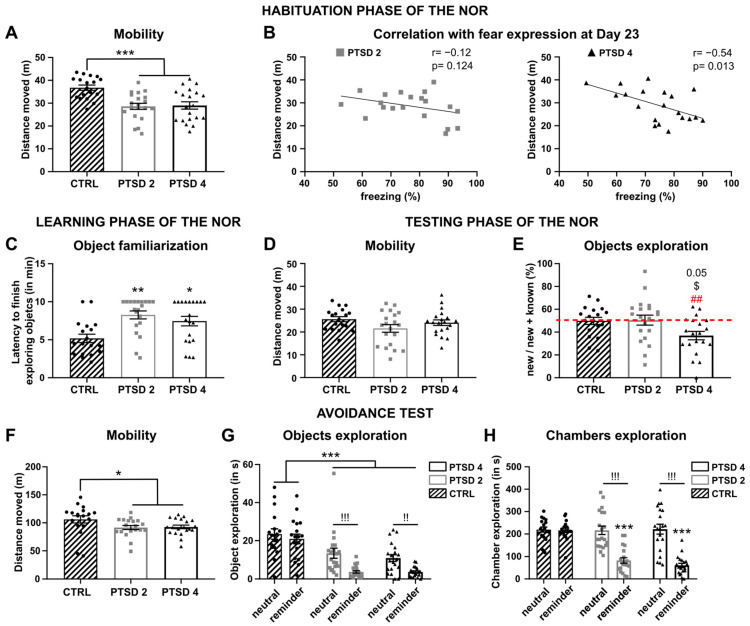
The long-term effect of the number of footshocks on PTSD-like behaviors before extinction learning. (**A**) The distance traveled in the arena during the habituation phase of the NOR was deceased in the conditioned group compared to CTRL, two weeks after fear conditioning. The arena was an unknown circular open field, dimly lit. (**B**) This distance traveled was negatively correlated to freezing expressed at the first re-exposure only for PTSD4 mice. (**C**) During the learning phase of the NOR, both conditioned groups took longer to become familiarized with the object compared to CTRL. (**D**) During the testing phase of the NOR, all groups traveled the same distance within the open field and (**E**) CTRL mice did not explore a new object more compared to the known object, underlying an absence of learning. However, PTSD4 mice explored this new object less compared to the known object (<50%), the CTRL group (*p* = 0.05) and the PTSD2 group. (**F**) Both fear-conditioned groups traveled less distance during the avoidance test compared to CTRL. The arena was divided in three equal chambers and placed under a low red light. (**G**) The neutral object corresponded to a known object previously displayed in the home cage. The reminder was the object that was present during fear conditioning. CTRL mice explored both object the same amount of time and significantly more than conditioned mice. PTSD2 and PTSD4 mice explored the reminder significantly less compared to the neutral object. (**H**) The objects were placed in the two opposite chambers and time spent in each chamber was measured. Conditioned mice spent significantly less time in the chamber containing the reminder compared to the other chamber or CTRL no matter the number of footshocks. Results are represented by the mean ± SEM, *** *p* < 0.001, ** *p* < 0.01 and * *p* < 0.05 vs. CTRL; !! *p* < 0.01 and !!! *p* < 0.001 neutral object vs. reminder within the same group, ## *p* < 0.01 vs. 50%, ^$^
*p* < 0.05 vs. PTSD2. CTRL, control group (*n* = 18); PTSD2, mice that had received two shocks (*n* = 20); PTSD4, mice that had received four shocks (*n* = 20); NOR, Novel Object Recognition test.

**Figure 4 brainsci-14-00311-f004:**
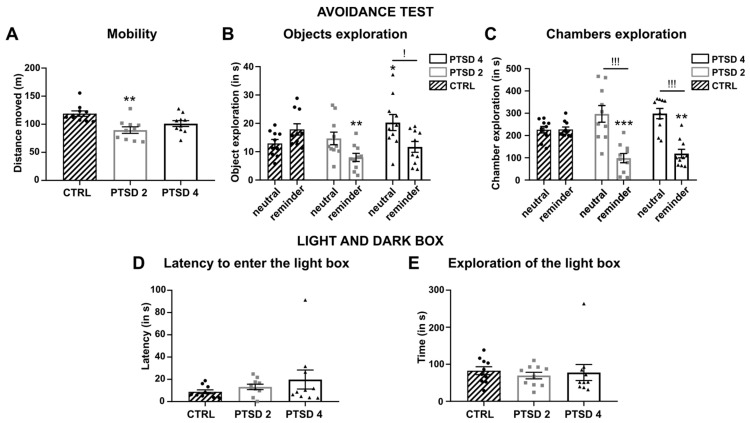
Maintenance of avoidance behaviors after extinction. (**A**) PTSD2 mice traveled less distance during the avoidance test after extinction compared to CTRL mice. (**B**) PTSD2 mice explored the reminder significantly less compared to CTRL. However, discrimination between the reminder and the neutral object was better in the PTSD4 group resulting in a significantly shorter exploration of the reminder compared to the neutral object. (**C**) Conditioned mice from both groups explored the chamber containing the reminder significantly less compared to the chamber containing the neutral object and compared to CTRL. (**D**) The latency to enter the illuminated box during a light and dark box test (LDB) was not modified in any conditioned group compared to CTRL. (**E**) The time spent within the illuminated box was not different in any conditioned group compared to CTRL. Results are represented by the mean ± SEM, * *p* < 0.05, ** *p* < 0.01 and *** *p* < 0.001 compared to CTRL; ! *p* < 0.05 and !!! *p* < 0.001 neutral object vs. reminder within the same group. CTRL, control group (*n* = 10); PTSD2, PTSD-like group fear conditioned with two footshocks (*n* = 10); PTSD4, PTSD-like group fear conditioned with four footshocks (*n* = 10).

**Table 1 brainsci-14-00311-t001:** Statistical result corresponding to the extinction acquisition within the three re-exposures to the conditioning context.

RM GLM Interactions *	*p*-Value	Post hoc Comparisons	Interpretation	*p*-Value
Day × Time × group: F(12, 330) = 1.28	*p* = 0.2261	-	-	-
Day × group: F(4, 110) = 21.22	***p* < 0.0001**	Extinction acquisition between sessions within the CTRL group	day 1 = day 2	*p* = 0.8741
			day 2 = day 3	*p* > 0.9999
		Extinction acquisition between sessions within the PTSD 2 group	**day 1 > day 2**	***p* < 0.0001**
			**day 2 > day 3**	***p* = 0.0002**
		Extinction acquisition between sessions within the PTSD 4 group	**day 1 > day 2**	***p* < 0.0001**
			**day 2 > day 3**	***p* = 0.0006**
		Extinction acquisition differences at day one between groups	**CTRL < PTSD 2**	***p* < 0.0001**
			**CTRL < PTSD 4**	***p* < 0.0001**
			PTSD 2 = PTSD 4	*p* = 0.9320
		Extinction acquisition differences at day two between groups	**CTRL < PTSD 2**	***p* < 0.0001**
			**CTRL < PTSD 4**	***p* < 0.0001**
			PTSD 2 = PTSD 4	*p* = 0.9925
		Extinction acquisition differences at day three between groups	**CTRL < PTSD 2**	***p* < 0.0001**
			**CTRL < PTSD 4**	***p* < 0.0001**
			PTSD 2 = PTSD 4	*p* = 0.8312
Time × group: F(6, 165) = 15.82	***p* < 0.0001**	Extinction acquisition within sessions for the PTSD 2 group	**5 min > 15 min**	***p* < 0.0001**
			**10 min > 20 min**	***p* < 0.0001**
		Extinction acquisition within sessions for the PTSD 4 group	**5 min > 15 min**	***p* < 0.0001**
			**10 min > 20 min**	***p* < 0.0001**

* A repeated measures general linear model (RM GLM) was used and post hoc tests were conducted on significant interactions (*p* > 0.05). Day corresponds to the re-exposure sessions and time to the 5 min windows within a re-exposure session. In bold are the significant results.

## Data Availability

The data presented in this study are openly available on Zenodo: Eyraud, N., Bloch, S., Brizard, B., Pena, L., Tharsis, A., Surget, A., El-Hage, W., & Belzung, C. (2024). Row Data for the paper: Influence of Stress Severity on Contextual Fear Extinction and Avoidance in a Posttraumatic-Like Mouse Model [Data set]. Zenodo. https://doi.org/10.5281/zenodo.10867894.

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
