# Peer review of "Influence of Stress Severity on Contextual Fear Extinction and Avoidance in a Posttraumatic-like Mouse Model"

_brainsci, 2024, doi:10.3390/brainsci14040311_

Round 1

Reviewer 1 Report

Comments and Suggestions for Authors

PTSD is recognized to be a development of long-lasting symptoms following exposure to life threatening experience. Although a substantial body of research has greatly improved our knowledge regarding prevalence, clinical symptoms and consequences of PTSD over the last decades, relatively little is known about underlying neurobiological abnormalities. This is why various animal models have been proposed. Current therapy is based on combining psychotherapy (e.g. exposure therapy) and pharmacotherapy (e.g. paroxetine). Unfortunately, both methods have drawbacks and are not effective in many patients. The authors of the study attempted to modify one of the known animal models of PTSD - fear conditioning, so that it could facilitate the mimicry of exposure therapy. The aim of the study was also the investigation the impact of varying the severity of aversive stimuli during the contextual fear conditioning procedure on the strength and the persistence of PTSD behavioural symptoms in mice. Overall this a well-designed study and provides. the article contains an interesting discussion about the fear conditioning model. However, in my opinion the significance of this manuscript within its field is low. The obtained results confirm previous reports on the possible ceiling effect of increasing the number of shocks on fear reaction to a conditioned context. I suggest that the Authors clarify the sentences summarizing the results - this model could be useful to study new therapeutic approaches to enhance extinction learning, and to alleviate PTSD symptoms such as fear generalization and trauma-related avoidance.

Author Response

Thank you very much for taking the time to review this manuscript and for pointing out a needed clarification at the end of the manuscript. Accordingly, we added a few lines at the end of the conclusion. Please find the detailed responses below and the corresponding revisions highlighted in the re-submitted file.

Modifications were made in the 5. conclusion paragraph p.14, lines 582 to 587:

Indeed, we know that extinction learning mostly relies on the top-down control of the vmPFC over the amygdala. Using this model to study new therapeutic approaches that could aim to increase vmPFC activity (e.g. neurostimulation techniques [48,56–58]) or enhance fear extinction with molecules known to interact with this inhibitory learning (e.g. curcumin [59,60] or ketamine [61]) might help us find efficient protocols to accompany or even enhance trauma-focused psychotherapies.

Reviewer 2 Report

Comments and Suggestions for Authors

·         The article mentions the complexity and limitations of current PTSD treatments, indicating a need for improvement. However, it could benefit from a more detailed exploration of alternative or adjunctive therapies to provide a broader perspective on treatment options.

·         While discussing the use of animal models for PTSD, the introduction could emphasize the inherent limitations of translating findings from animal studies to human conditions, considering the complex nature of PTSD in humans which involves cognitive, emotional, and social components that might not be fully replicated in animals.

·         The manuscript should provide further clarification regarding why there was a reduction in the number of mice tested after extinction compared to before extinction for the same avoidance test (n=10).

·         Provide a clear rationale for the chosen experimental design, including the selection of animal models, experimental conditions, and behavioral assays. Justify any deviations from standard protocols and explain their relevance to the research question.

·         Attached are two articles from multiple which present how PTSD or fear are not exhibited the same among males and females. In your study, you only used male mice (which can be considered a limitation). Please mention this limitation and other limitations in this section.

Shansky. https://doi.org/10.1016/j.ynstr.2014.09.005

·         Provide more in-depth interpretations of the results, particularly regarding the implications of the observed fear responses, extinction learning, and PTSD-like behaviors.

·         Discuss possible mechanisms underlying the observed phenomena, such as the role of fear generalization and the impact of stress severity on avoidance behaviors. Suggested article- http://dx.doi.org/10.1016/j.biopsych.2015.04.010

·         Examining outcomes at later timepoints greater than one month after conditioning/extinction may better capture the persistence of effects. Using distinct contextual generalization testing could help differentiate associative vs non-associative components.

·         At the end it is mentioned how some of the mice did maintain PTSD-like behavior. I think using those mice for additional studies such as studying the effect of anti-stress drugs (like Curcumin) or treating them to reduce their PTSD behavior and decreasing their stress would be helpful. Attached is a couple of articles exhibiting the effect of curcumin-based compounds on certain brain receptors (like AMPA receptors) and the role of this compound in

Qneibi et al. https://doi.org/10.1016/j.bioorg.2021.105406

Author Response

Thank you very much for taking the time to review this manuscript and the different comments you have made to help us improve it. Please find the detailed responses below and the corresponding corrections highlighted in the re-submitted files. 

Comments 1: The article mentions the complexity and limitations of current PTSD treatments, indicating a need for improvement. However, it could benefit from a more detailed exploration of alternative or adjunctive therapies to provide a broader perspective on treatment options.

Response: Thank you for pointing this out. A paragraph was added at the beginning of the Introduction to mention some pharmacological or neurostimulation technics that could be used. Modifications were made to lines 35 to 52:

Additionally, only a few antidepressant medications can be prescribed, especially selective serotonin reuptake inhibitors (SSRI) [7,8]. However, even when combined with psychotherapy, the therapeutic outcomes are relatively weak and not as recommended as psychotherapy alone [7,9]. Therefore, there is a pressing need to discover faster and more efficient methods to treat the condition and alleviate patients from the burden of lengthy, distressing or ineffective treatments. While some have attempted to develop new pharmacological treatments (e.g. propranolol, D-cycloserine, oxytocin …)[10], others have explored non-pharmacological and non-invasive methods to alleviate brain activity dysregulations found in PTSD. Numerous studies have thus focused on developing and adapting neurostimulation techniques to treat PTSD and augment the effects of psychotherapies [11–14]. Although results are promising, discrepancies in outcomes and methodologies hinder drawing definitive conclusions about the efficacy of these methods, particularly repeated neurostimulation. Despite the challenge of fully capturing all dimensions of the disease and its complexity in animal models, there is a genuine need for additional preclinical studies to systematically explore new therapeutic avenues and elucidate the mechanisms underlying successes or failures of treatments. This is why we aimed to refine and adjust a rodent model to better represent the disorder and simulate exposure therapy.

Comments 2: While discussing the use of animal models for PTSD, the introduction could emphasize the inherent limitations of translating findings from animal studies to human conditions, considering the complex nature of PTSD in humans which involves cognitive, emotional, and social components that might not be fully replicated in animals.

Response: We added the following into the Introduction and Discussion. Modifications were made to lines 47 to 51:

Despite the challenge of fully capturing all dimensions of the disease and its complexity in animal models, there is a genuine need for additional preclinical studies to systematically explore new therapeutic avenues and elucidate the mechanisms underlying successes or failures of treatments.

Modifications were also made to lines 562 to 564:

Additionally, this underscores the challenge of translating animal models to humans, despite the necessity of animal investigation to explore new avenues in clinical research.

Comments 3: The manuscript should provide further clarification regarding why there was a reduction in the number of mice tested after extinction compared to before extinction for the same avoidance test (n=10).

Response: We agree with this comment. Hence, an explanation was added to the Methods 2.2 Experimental design. Modifications were made to lines 136 to 140:

The experiment was initially conducted with 28 mice (first batch) and repeated with an additional 30 mice (second batch) with a two-week interval. Only the second batch underwent investigation for avoidance and anxiety behaviors following extinction learning, with 10 mice per group. The first batch was euthanized for a histological investigation, which was unrelated to the present study.

Comments 4: Provide a clear rationale for the chosen experimental design, including the selection of animal models, experimental conditions, and behavioral assays. Justify any deviations from standard protocols and explain their relevance to the research question.

Response: We added some clarifications throughout the manuscript, beginning with an elucidation of the limitations of current treatments in reference to the initial comment. Additionally, we included a few lines explaining the significance of eliciting PTSD-like behaviors after extinction training and why we specifically opted to increase the number of shocks.  

Modifications were made to lines 97 to 105:

However, besides fear generalization, few PTSD-like symptoms have been observed to manifest following an extinction training. If future studies aim to explore the advantages of augmenting extinction learning, a model showing symptoms after extinction training may prove beneficial.   

PTSD has frequently been conceptualized as a robust fear conditioning learning that is resistant to extinction-inhibitory processes. However, PTSD is also thought to be influenced by the severity of the trauma and the level of arousal during fear acquisition [9]. Therefore, the greater the fear and stress levels experienced during the event, the higher the risk of developing PTSD.

Comments 5: Attached are two articles from multiple which present how PTSD or fear are not exhibited the same among males and females. In your study, you only used male mice (which can be considered a limitation). Please mention this limitation and other limitations in this section.

Response: Agree. A paragraph at the end of the discussion was added discussing some limitations of this study. Modifications were made to lines 552-571:

This study has several limitations that should be acknowledged. Firstly, the most significant limitation is its restrictions to male subjects. PTSD has been found to occur twice as often in women compared in men. Thus, it is crucial to include females in preclinical and fundamental studies. However, it is worth noting that the type of traumatic event could also play a significant role. For instance, sexual assaults have been particularly associated with the development of PTSD, a type of trauma more often experienced by women [2,52]. Furthermore, a few studies on contextual fear conditioning have highlighted differences between males and females in the acquisition and expression of fear extinction, showing a higher level of freezing in males compared to females. However, this suggest that freezing might not be the sole representation of fear, and other behaviors must be defined and refined [53]. Additionally, this underscores the challenge of translating animal models to humans, despite the necessity of animal investigations to explore new avenues in clinical research. Moreover, post-extinction behaviors were assessed one month after CFC. One month is already a lengthy period for fear conditioning studies in rodents, typically limited to 28 days. Nonetheless, it could be valuable to continue this investigation at other time points to capture long-term effects of extinction. Finally, a group subjected to fear conditioning with a protocol not expected to induce PTSD-like behaviors (e.g., a single 0.7 mA shock) could have been useful to discern what is attributable to fear learning and what can genuinely be associated with extreme fear consequences.

Comments 6: Provide more in-depth interpretations of the results, particularly regarding the implications of the observed fear responses, extinction learning, and PTSD-like behaviors.

Response: Few sentences were added to complete the discussion, but we mainly discussed it in relation to fear generalization and avoidance (see next comment corrections).

Modifications were made to lines 443 to 446:

When the first re-exposure began, mice from both PTSD groups froze around 75% of the time at the beginning of the session, suggesting a significant association of the context with the fearful event

Lines 453 to 456:

This could imply that a maximum level of freezing (ceiling effect) is attained after two or more shocks, event weeks after the conditioning acquisition, when long-term memory has been fully stored and consolidated.

Lines 467 to 471:

Hence, it appears that increasing the number of shocks did not affect extinction learning nor did it enhance conditioning memory. There does not seem to be a differential interpretation of threat severity between two or four shocks. Nonetheless, fear expression remained high even after a three-day extinction protocol. Although the extinction training was not entirely ineffective, there is still room for improvement.

Lines 527 to 529 :

We observed a significant avoidance of a reminder of the conditioned context that persisted even after an extinction protocol, providing further evidence extinction learning was incomplete

Comments 7: Discuss possible mechanisms underlying the observed phenomena, such as the role of fear generalization and the impact of stress severity on avoidance behaviors. Suggested article- http://dx.doi.org/10.1016/j.biopsych.2015.04.010

Response: Thank you for this comment and the suggestion of reference. We related fear generalization to the infralimbic cortex which correspond anatomically and functionally to the vmPFC in humans. Modifications were made to:

Lines 515 to 523:

Fear generalization after CFC has been shown to depend on the infralimbic cortex (IL- the rodent equivalent of the vmPFC) [46], just like the vmPFC seems implicated in fear generalization in humans [47]. Additionally, the IL plays a central role in extinction learning and expression [48]. Studies have revealed that CFC leads to a reduction in IL excitability [49]. Thus, fear generalization and only partial extinction learning may reflect a long-term impact of the stressful event on the IL functioning potentially contributing to PTSD-like symptoms. Further exploration on the IL’s involvement in overgeneralization and specific avoidance shed light on the efficacy of targeting fear extinction in PTSD treatment.

Lines 544 to 547:

Therefore, this avoidance behavior does not indicate a generalization of fear to other contexts but a specific association of the fearful event with the contextual elements during conditioning, which could potentially be ameliorated through enhanced extinction learning.

Comments 8: Examining outcomes at later timepoints greater than one month after conditioning/extinction may better capture the persistence of effects. Using distinct contextual generalization testing could help differentiate associative vs non-associative components.

Response: Indeed, later timepoints could be interesting and very useful. However, a PTSD-like model in rodent cannot represent all the complexity of this human disorder and having behaviors maintained up to one month is already challenging. We needed to first make sure that extinction training would not alleviate those behaviors. Specific avoidance had been shown to generalize to other cues with time (see Pamplona et al (2011), 28 days vs 1 day post conditioning). Now that we found a protocol that maintains specific avoidance after 28 days and even after extinction, testing at later timepoints could be interesting.

Modifications were made to:

Lines 500 to 507:

Testing fear generalization in contexts slightly similar to the conditioned one could have helped distinguish associative from non-associative type of generalizations (e.g. using a context with a grid floor and another similar one in terms of size), potentially enhancing the assessment of overgeneralization by measuring freezing expression [31–33]. However, it is essential to note that not measuring the freezing response directly but rather exploring other behaviors should be more systematically considered, as emphasized by the results of this study. Freezing is indeed a specific indicator of fear but considering other less obvious behaviors might help prevent under-interpretation.

Lines 564 to 567:

Moreover, post-extinction behaviors were assessed one month after CFC. One month is already a lengthy period for fear conditioning studies in rodents, typically limited to 28 days. Nonetheless, it could be valuable to continue this investigation at other time points to capture long-term effects of extinction.

Comments 9: At the end it is mentioned how some of the mice did maintain PTSD-like behavior. I think using those mice for additional studies such as studying the effect of anti-stress drugs (like Curcumin) or treating them to reduce their PTSD behavior and decreasing their stress would be helpful. Attached is a couple of articles exhibiting the effect of curcumin-based compounds on certain brain receptors (like AMPA receptors) and the role of this compound in Qneibi et al. https ://doi.org/10.1016/j.bioorg.2021.105406

Response: Thanks again for the proposed reference. we clarified how this model could be useful for future study at the end of the conclusion.

Modifications were made to:

Lines 581 to 587

Indeed, we know that extinction learning mostly relies on the top-down control of the vmPFC over the amygdala. Using this model to study new therapeutic approaches that could aim to increase vmPFC activity (e.g. neurostimulation techniques [48,56–58]) or enhance fear extinction with molecules known to interact with this inhibitory learning (e.g. curcumin [59,60] or ketamine [61]) might help us find efficient protocols to accompany or even enhance trauma-focused psychotherapies.

Reviewer 3 Report

Comments and Suggestions for Authors

The Authors have presented the research carried out in great detail, although slightly chaotic in some places. The following is a list of comments:

1. In the introduction, the authors focused on one treatment for PTSD and only slightly mentioned pharmacotherapy. What is missing here is a broader description of coping with post-traumatic stress. And an indication and greater emphasis on why this treatment model was chosen? What is the rationale for its use, although as they themselves mentioned it involves frequent drop-out.

2. Lines 176-177 - what was the object of the 'reminder'?

3. Lines 178-179 similarly a 'neutral object'?

4. Lines 190-191 - this sentence is very confusing. When did the mice finally find themselves in the behavioural room? Needs to be rewritten

5. Section 2.4.4 - I assume that there was a door between the boxes through which the mice could roam freely

6. Line 214 - here again the term" behavioural room" appears, which box performed such a role?

7. All statistics are presented in supplementaries. It would be good to rearrange those that indicate statistically significant differences in table form, but also in the main text. This will be more readable and not refer the reader to the supplementary material.

Author Response

Thank you very much for taking the time to review this manuscript and for your suggestions to help us improve it. Please find the detailed responses below and the corresponding revisions highlighted in the re-submitted files. 

  1. In the introduction, the authors focused on one treatment for PTSD and only slightly mentioned pharmacotherapy. What is missing here is a broader description of coping with post-traumatic stress. And an indication and greater emphasis on why this treatment model was chosen? What is the rationale for its use, although as they themselves mentioned it involves frequent drop-out.

Response: Thank you for this comment. We added a few lines at the end of the conclusion.

Lines 581 to 587

Indeed, we know that extinction learning mostly relies on the top-down control of the vmPFC over the amygdala. Using this model to study new therapeutic approaches that could aim to increase vmPFC activity (e.g. neurostimulation techniques [48,56–58]) or enhance fear extinction with molecules known to interact with this inhibitory learning (e.g. curcumin [59,60] or ketamine [61]) might help us find efficient protocols to accompany or even enhance trauma-focused psychotherapies.

  1. Lines 176-177 - what was the object of the 'reminder'?

Response: Both objects used during the avoidance tests were clarified to make it clearer for the readers. The reminder is an object that was present during the conditioning as part of the surrounding context.

Lines 200 to 203:

The object present during conditioning (the glass prism - 2.5x2.5x2.5x6.5 cm- which we referred as the reminder) was exhibited in one of the chambers adjacent to central chamber and cleaned with the same Eugenol solution.

  1. Lines 178-179 similarly a 'neutral object'?

Response: Lines 203 to 206:

On the opposite chamber, cleaned with water, a neutral object (a flat and large metallic ring - 6x1(h) cm with a 3-cm-large hole in the middle) to which animals were habituated in their home-cage before the test was first displayed (4 min and 36 sec on day 20).

  1. Lines 190-191 - this sentence is very confusing. When did the mice finally find themselves in the behavioural room? Needs to be rewritten

Response: Agreed, we did not employ a clear vocabulary to differentiate the apparatus used to test the behaviors and the room of the facility in which we worked. Hence, the behavioral room is not a test apparatus but a room of the facility that was dedicated to behavioral tests. To clarify and avoid confusions, the word “room“ was employed only to talk about this behavioral room. Otherwise, we employed “chamber” or “box” to talk about specific compartments of an apparatus.  

Modifications:

Lines 216 to 218:

The Mice were transferred from their housing room to the behavioral testing room 30 min. before initiating the avoidance tests, allowing them to acclimate to the new environment.

  1. Section 2.4.4 - I assume that there was a door between the boxes through which the mice could roam freely

Response: Indeed, thank you for pointing this out. Modifications were made in this regard. Lines 237 to 238:

A small opening enabled the mice to freely explore one box or the other.

  1. Line 214 - here again the term" behavioural room" appears, which box performed such a role?

Response: Modifications were made from line 241 to 244

Thirty minutes before the test, all mice were transferred to the behavioral testing room to allow them to acclimate to the environment and to mitigate any stress biases associated with the relocation.  

  1. All statistics are presented in supplementaries. It would be good to rearrange those that indicate statistically significant differences in table form, but also in the main text. This will be more readable and not refer the reader to the supplementary material.

Response: To avoid information overload and the GLM results being the only results not clearly represented by stars, exclamation points or hashtags on the figures, a table was added in the main text only for this statistical analyze. You will find it in the new manuscript page 8.

Table 1. Statistical result corresponding to the extinction acquisition within the three re-exposures to the conditioning context.

RM GLM interactions *

p-value

post-hoc comparisons

interpretation

p-value

Day×Time×group:

F(12, 330)=1.28

p=0.2261

-

-

-

Day×group: F(4, 110)=21.22

p<0.0001

Extinction acquisition between sessions within the CTRL group

day 1 = day 2

p=0.8741

day 2 = day 3

p>0.9999

Extinction acquisition between sessions within the PTSD 2 group

day 1 > day 2

p<0.0001

day 2 > day 3

p=0.0002

Extinction acquisition between sessions within the PTSD 4 group

day 1 > day 2

p<0.0001

day 2 > day 3

p=0.0006

Extinction acquisition differences at day one between groups

CTRL < PTSD 2

p<0.0001

CTRL < PTSD 4

p<0.0001

PTSD 2 = PTSD 4

p=0.9320

Extinction acquisition differences at day two between groups

CTRL < PTSD 2

p<0.0001

CTRL < PTSD 4

p<0.0001

PTSD 2 = PTSD 4

p=0.9925

Extinction acquisition differences at day three between groups

CTRL < PTSD 2

p<0.0001

CTRL < PTSD 4

p<0.0001

PTSD 2 = PTSD 4

p=0.8312

Time×group: F(6, 165)=15.82

p<0.0001

Extinction acquisition within sessions for the PTSD 2 group

5min > 15min

p<0.0001

10min > 20min

p<0.0001

Extinction acquisition within sessions for the PTSD 4 group

5min > 15min

p<0.0001

10min > 20min

p<0.0001

* A repeated measures general linear model (RM GLM) was used and post-hoc tests were conducted on significant interactions (p>0.05). Day correspond to the re-exposures sessions and Time to the 5-min-windows within a re-exposure session. In bold are the significant results.

Round 2

Reviewer 2 Report

Comments and Suggestions for Authors

Thank you for taking all comments into consideration. The manuscript has improved significantly.